# Characterization of Zr-Containing Dispersoids in Al–Zn–Mg–Cu Alloys by Small-Angle Scattering

**DOI:** 10.3390/ma16031213

**Published:** 2023-01-31

**Authors:** Mohammad Taha Honaramooz, Roland Morak, Stefan Pogatscher, Gerhard Fritz-Popovski, Thomas M. Kremmer, Thomas C. Meisel, Johannes A. Österreicher, Aurel Arnoldt, Oskar Paris

**Affiliations:** 1Chair of Physics, Department Physics, Mechanics and Electrical Engineering, Montanuniverstät Leoben, Franz-Josef-Str. 18, 8700 Leoben, Austria; 2R&D Material Science CMI, Center for Material Innovation, AMAG Rolling GmbH, Postfach 32, 5282 Ranshofen, Austria; 3Chair of Nonferrous Metallurgy, Department Metallurgy, Montanuniverstät Leoben, Franz-Josef-Str. 18, 8700 Leoben, Austria; 4Chair of General and Analytical Chemistry, Department General, Analytical and Physical Chemistry, Montanuniverstät Leoben, Franz-Josef-Str. 18, 8700 Leoben, Austria; 5LKR Light Metals Technologies, Austrian Institute of Technology, Lamprechtshausenerstraße 61, 5282 Ranshofen, Austria

**Keywords:** SAXS, SANS, Zr-containing dispersoid, Al–Zn–Mg–Cu, size distribution

## Abstract

The characterization of Zr-containing dispersoids in aluminum alloys is challenging due to their broad size distribution, low volume fraction, and heterogeneous distribution within the grains. In this work, small-angle X-ray scattering (SAXS) and small-angle neutron scattering (SANS) were compared to scanning electron microscopy (SEM) and transmission electron microscopy (TEM) regarding their capability to characterize Zr-containing dispersoids in aluminum alloys. It was demonstrated that both scattering techniques are suitable tools to characterize dispersoids in a multi-phase industrial 7xxx series aluminum alloy. While SAXS is more sensitive than SANS due to the high electron density of Zr-containing dispersoids, SANS has the advantage of being able to probe a much larger sample volume. The combination of both scattering techniques allows for the verification that the contribution from dispersoids can be separated from that of other precipitate phases such as the S-phase or GP-zones. The size distributions obtained from SAXS, SANS and TEM showed good agreement. The SEM-derived size distributions were, however, found to significantly deviate from those of the other techniques, which can be explained by considering the resolution-limited restrictions of the different techniques.

## 1. Introduction

Age-hardenable 7xxx series aluminum alloys are used in the aerospace industry due to their high specific strength and good stress corrosion cracking (SCC) resistance [1]. However, thick plates from these alloys are prone to having a non-uniform microstructure throughout their thickness due to different levels of deformation and, as a result, different degrees of recrystallization [2]. To deal with this issue, transition elements are added to form precipitates, so-called dispersoids, which inhibit/retard recrystallization [3,4,5]. Regarding quench-sensitivity, Cr- and Mn-containing dispersoids are incoherent and act as preferential sites for the nucleation of the quench-induced η-phase precipitates [6]. On the other hand, Zr-containing dispersoids can provide a high coherency with the matrix and show a lower tendency to trigger the heterogeneous nucleation of the precipitates if they have a coherent L1_2_ structure [2,7]. Moreover, Zr-containing dispersoids show a high thermal stability, making them ideal for high-temperature applications [8].

The Al–Zr phase diagram exhibits a peritectic solidification and an equilibrium partition coefficient higher than one. In other words, the Zr distribution is not uniform, and there is a gradient of Zr concentration decreasing from the dendritic cells toward the interdendritic channels [9]. This directly results in the non-uniform distribution of Zr-containing dispersoids within the grains [3]. Furthermore, it has been shown that Zr concentration influences the nucleation rate, temperature, and size of the dispersoids [9,10,11,12]. Dispersoid size depends on several factors, such as chemical composition and heat-treatment parameters. Therefore, a broad range of sizes is reported in the literature [9,11,13,14]. The most common technique for investigating dispersoids is transmission electron microscopy (TEM), which directly reveals their size (distribution), shape, and mutual arrangement and delivers information about their coherency. However, one of the most significant drawbacks of TEM is that a limited number of dispersoids is included in the analysis because the analyzed volume is very small [9,15,16,17], which is particularly critical if the distribution of the precipitates is not homogeneous.

The small-angle scattering (SAS) of X-rays (SAXS) or neutrons (SANS) is an alternative technique that provides information about the size, size distribution, volume fraction, and morphology of precipitates from a considerably larger sampling volume [18,19]. Although these techniques cannot uniquely distinguish between different types of precipitates in multi-phase alloys, they have their strengths, particularly for binary systems [20]. So far, multiphase Al alloys have been primarily investigated with SAS regarding aging (natural or artificial) [21,22,23,24,25], but very few works have considered SAS in studying Zr-containing dispersoids [26,27]. SAS covers a much larger sample volume, thus providing better statistics and averages over heterogeneous regions, while electron microscopy methods directly visualize the precipitates. Thus, combining microscopy and scattering techniques allows us to obtain a much more comprehensive characterization of Zr-containing dispersoids.

This work combines SAXS and SANS to study dispersoids in a cast and homogenized industrial alloy in its full compositional and processing complexity. The different sensitivities of X-rays and neutrons to the chemical elements in the alloy are employed to estimate the influence of other precipitate types such as the η-phase or GP-zones. The dispersoid-size distributions obtained from SAXS and SANS are compared to the distributions from the TEM and SEM of the same samples, and the capability of SAS to quantitatively characterize low-volume fractions of dispersoids in complex multi-phase Al-alloys is validated.

## 2. Materials and Methods

A commercial direct-chill (DC) cast Al–Zn–Mg–Cu–Zr alloy was used in this work. Samples were cut out from the as-cast bar with dimensions of 6 × 1.5 × 0.45 m^3^, covering one-quarter of the bar cross-section on a regular grid (7 × 3 = 21 samples, see Figure 1).

These samples were subsequently homogenized and quenched into water. Table 1 reports the average chemical composition of the alloy over the whole quadrant measured by both X-ray fluorescence analysis (XRF) and ICP OES. All XRF measurements were conducted on a clean sample surface (cleaned and ultrasonicated with ethanol) with a wavelength dispersive XRF instrument (AXIOS Malvern Panalytical), applying a 10 mm mask. The peak evaluation program (Ominan10) with a semiquantitative calibration based on fundamental parameters using single element calibration standards was used for evaluation.

Small-angle neutron scattering (SANS) measurements were conducted on 21 rectangular sample sheets, prepared from the above samples with dimensions of 35 × 23.5 × 1 mm^3^, using the D33-massive dynamic q-range small-angle diffractometer-instrument at the Institute Laue-Langevin (ILL) in Grenoble (France). The beam size was 15 × 15 mm, and a neutron wavelength (λ) of 5.3Å was chosen to prevent double Bragg diffraction. Two sample-to-detector distances were employed to cover the total scattering vector length q from 0.06 nm^−1^ to 3.7 nm^−1^ ($q=4\pi sin\left(\theta \right)/\lambda$*,* with 2θ being the scattering angle). The 2D SANS patterns measured with an area detector were corrected for background, transmission, and sample thickness, and then were azimuthally averaged. The incoherent scattering of a water sample was used to calculate the absolute value of the differential scattering cross-section.

Small-angle X-ray scattering (SAXS) samples with the dimensions (20 × 20) mm^2^ and a thickness of (100 ± 20) µm were prepared from regions adjacent to the SANS samples by grinding and polishing. Cu K_α_ X-ray radiation (λ = 0.154 nm) with a beam diameter of 0.5 mm was used. The sample-to-detector distance was 67 cm, leading to a somewhat smaller *q*-range of 0.09 nm^−1^ to 3.6 nm^−1^ compared to SANS. To cover a comparable sample area with SAXS, 25 measurement points were selected on a 5 × 5 regular grid with 1 mm distance for each sample, and each point was exposed to the X-ray beam for 20 min. Double Bragg diffraction and total reflection from grain boundaries occasionally showed up in the 2D SAXS patterns, recognizable by single spots or sharp streaks. These features were carefully masked for each pattern. The SAXS pattern of each individual measurement point was then azimuthally averaged and corrected for time, transmission, and background, and an average SAXS curve was calculated based on all 25 measurements. The size distribution of the dispersoids was calculated from the scattering profiles using the model described in Section 3.1 and Appendix A.

Scanning electron microscopy (SEM) examination was conducted on six samples (see Figure 1), prepared from regions close to the corresponding SAXS/SANS samples after polishing in up to 1 µm of OPS solution. For the quantification of dispersoids, backscattered electron (BSE) images were acquired using a TESCAN Mira 3 microscope equipped with a four-quadrant BSE detector at an accelerating voltage of 20 kV and a working distance of 10 mm, resulting in a sample-to-detector distance of 3 mm. The images covered a field of view of approximately 93 µm and a resolution of 8192 × 6144 pixels. A micrograph of the edge region, a quarter of the plate thickness, and the middle was taken for each sample. For the segmentation of dispersoids, the free software ImageJ was used. The micrographs were first divided 8 by 8, resulting in 64 partial images each. Only partial images not containing coarse primary intermetallic phases were used for segmentation because coarse phases make it difficult to set a threshold value. Then, manual thresholding was performed separately using the triangle algorithm for each partial image. Particles with a pixel size of 5 to 60 and circularity of 0.4 to 1.0 were counted as dispersoids (between 2000 and 7000 for each sample position). The circle-equivalent radii of the particles were used for creating areal size distributions. Using this procedure, large areas could be evaluated.

Sample preparation for TEM was conducted by punching 3 mm discs from the mechanically pre-polished SAXS foils. Subsequently, twin-jet polishing by a Struers TenuPol 5 system with a voltage between 13 V and 15 V at a temperature of −22 °C was carried out until an electron-transparent area was obtained. A 67% methanol/33% HNO_3_ electrolyte was used for the electropolishing step. The TEM investigations included scanning transmission electron microscopy high-angle annular darkfield (STEM-HAADF), electron diffraction, and TEM brightfield measurements performed using an FEI Tecnai F20 G2 (ThermoFisher, Hillsboro, OR, USA) with an acceleration voltage of 200 kV. TEM images were taken typically 10 to 20 µm away from the grain boundaries to avoid heterogeneities close to the grain boundaries. In order to analyze the TEM images and calculate the size distribution of the dispersoids, ImageJ software was used. After plane-level correction and additional smoothing steps, particle size evaluation was accomplished semi-automatically by grayscale thresholding. A minimum area cutoff of 15 nm² was applied to suppress noise for the measured particles. Additionally, particles with a circularity parameter (4π × area/perimeter^2^) [28] lower than 0.85, which correspond to overlapping particles, were removed from the data set. The measured particle area was then converted to an equivalent area circular diameter (ECD) [29], defined as the diameter of a circle with the same area as the particle.

## 3. Results

### 3.1. SANS/SAXS

Despite their low statistics, one of the most apparent advantages of TEM and SEM is that they deliver real-space images of precipitates. Therefore, their interpretation is much more straightforward, and they are more comprehensible than SAS results. Nevertheless, if a simple two-phase model (i.e., a single type of precipitates in a homogeneous matrix) is applicable, the average structural parameters such as the volume fraction, surface area, and average size of the precipitates are easily available from the integral parameters of the SAS curves [30]. Moreover, analytical or numerical models can be employed to fit the SAS data, revealing the size distribution as the main outcome if the shape of the precipitates is spherical or nearly spherical. Several research papers have provided comprehensive information on SAS data evaluation for metallic alloys [18,31,32]. However, several types of precipitates can be present in multicomponent industrial alloys, making SAS data analysis much more complicated and, in many cases, ambiguous. This is why we have chosen to combine SANS and SAXS, since the two probes exhibit different sensitivities for different precipitate types. In the following, we briefly present the data evaluation and interpretation used in this work, considering radially integrated and corrected SANS and SAXS data.

In Figure 2, representative SANS and SAXS data show the spherically averaged intensity *I* versus the length of the scattering vector *q* on a double-logarithmic scale. Since the SAXS data were not calibrated to absolute intensity, there is an unknown factor between the two data sets. In order to still compare them, the SAXS curve was multiplied with a constant factor, such that the two curves overlapped at the lowest *q*-values measured for SAXS. A power-law behavior with an exponent close to −4 is observed at a very low *q* for SANS (SAXS data are not available for a low enough *q*), which can be attributed to the very large particles (typically with micrometer extensions, such as the S-phase in the present case) [33]. Two characteristic humps are observed in the intermediate and large *q*-range, respectively. The hump at a large *q* with a maximum at about 2 nm^−1^ is interpreted to stem from Zn-rich GP-zones [21,34,35]. This scattering feature is well-separated from the rest of the curve, since the GP-zones are very small (a few nanometers only). The shoulder at an intermediate *q* (at about 0.2 nm^−1^) is attributed to Zr-containing dispersoids, as explained and justified below:

In principle, scattering of η-phase precipitates would be expected in a similar *q*-regime since their size is typically in the same order of magnitude as the one of the dispersoids. Yet, Priya et al. [36] reported that no η-phase should occur in the homogenized state. This means that the primary η-phase formed during solidification should be completely dissolved into the matrix during homogenization, and subsequent quenching of the samples into water should suppress the re-precipitation. Therefore, no η-phase should exist in the present samples [36,37]. This is consistent with the experimental observation that the SAXS and SANS curves for *q* < 0.3 nm^−1^ (i.e., in the region of the first hump in Figure 1) have an identical shape, which would not be expected if two different types of precipitates (i.e., dispersoids and η-phase) were present at the same time. This conclusion is also supported by the quantitative comparison of the scattering contrast for SANS and SAXS. As outlined in detail in the Appendix A, the SAXS intensity for the GP-zones (i.e., the SAXS signal at large *q*-values) should be smaller by roughly a factor of four compared to that of SANS, if the two curves are normalized to the same intensity for the dispersoids (i.e., in the intermediate *q*-range), irrespective of the volume fractions. This is exactly what is seen in Figure 2, which would not be the case if a noticeable amount of η-phase was present.

Therefore, for the quantitative evaluation of the SAXS and SANS data, we assumed that the measured intensity is approximated by the sum of three independent intensity contributions (see Appendix A). First, the scattering at a very small *q* is described by a *q*^−4^ power-law behavior, which is attributed to the large primary precipitates such as the S-phase. Second, the scattering from the GP-zones is described by a model based on scaling functions for phase-separating systems [38,39], which describes the SAS curves of non-diluted precipitate phases in alloys quite well. Since the GP-zones are expected to be considerably smaller than the dispersoids, their scattering contribution should be dominant only at a large *q*, i.e., in the case of *q* > 0.5 nm^−1^. Third, for the scattering from the dispersoids at an intermediate *q*, we employ a model for non-interacting spherical precipitates with a free-form size distribution, as described in detail in the Appendix A. Finally, a constant background scattering, arising mainly from Laue scattering due to the solid solution matrix, is also considered.

Figure 3a,b show examples of fitted curves for sample #13 for SANS and SAXS, respectively. Figure 3c,d show the resulting volume-normalized diameter distributions of the dispersoids obtained from these fits. The distribution obtained from SANS (Figure 3c) reveals a close similarity to that from SAXS (Figure 3d). For both, the maximum of the distribution is around 22 nm, and the minimum is around 12 nm. The cutoff at around 50 nm is largely determined by the resolution, i.e., by the lowest accessible *q* values. The only difference is that the size distribution from SAXS is somewhat broader for large diameters as compared to SANS. This can be attributed to the restricted *q*-range toward a low *q* for SAXS as compared to SANS, which is discussed in more detail in Section 4. The fitted dispersoid contribution to the scattering curves (green lines in Figure 3a,b) shows a first shoulder at around 0.2 nm^−1^ and a second shoulder at around 0.5 nm^−1^. The latter suggests a second class of dispersoid diameters around and below 10 nm, as indicated by the increase in size distributions toward small diameters in Figure 3c,d. We consider this an artifact from the non-perfect description of the GP-zones by the scaling model function, and, therefore, we do not interpret diameters below about 12 nm as belonging to the dispersoids.

### 3.2. Scanning Electron Microscopy (SEM)

Figure 4a,b show SEM images at different magnifications of sample #13 after homogenization and water quenching. Although precipitation occurred on the grain boundaries (Figure 4a), the volume fraction of these precipitates is small. Since the homogenization temperature was high enough to dissolve the η-phase, we can assume that these grain-boundary precipitates are S-phase. This confirms that, on the one hand, the S-phase remains after quenching the samples into water [40]. Therefore, a higher temperature or longer homogenization time is needed to eliminate the S-phase. On the other hand, a fast cooling rate after homogenization suppresses the η-phase re-precipitation, following the previous finding by Priya et al. [36]. Figure 4b displays a larger magnification of a grain in Figure 4a, clearly showing a dense distribution of fine precipitates that can be identified as dispersoids.

Figure 4c shows the size distribution of dispersoids extracted by SEM from sample #13. The dispersoid diameter starts from an interval of 25–30 nm (smaller sizes not being detectable) and ceases at about 100 nm. The average diameter (mean value of the fitted lognormal size distribution, as shown by the red curve) of the dispersoids calculated from the SEM size distributions is clearly higher than the calculated value from the SANS/SAXS data. This may be due to the fact that SAXS is sensitive to dispersoids as small as 10 nm in diameter, while SEM exhibits a resolution-limited cutoff at 25 nm. It should be noted that, in contrast to the SANS/SAXS distribution, the number density is presented in Figure 4.

### 3.3. Transmission Electron Microscopy (TEM)

Figure 5a–d show TEM images of four different samples, with spherical dispersoids of around 20–30 nm in diameter being clearly recognized. These values are in good agreement with the mean values obtained from SANS and SAXS. While, in some samples, these precipitates are finely dispersed (Figure 5a,c), they seem to form clusters in other samples (Figure 5b,d). For the evaluation of the size distribution, only regions within the grain interior, typically 10-20 µm from the grain boundaries (such as those shown in Figure 5), were considered. TEM images from regions closer to a grain boundary are shown in Appendix A for different samples. The number of dispersoids declines moving toward the grain boundary, and there is a tendency that their size increases. A largely precipitation-free zone (PFZ) close to the grain boundary was observed, which was already repeatedly reported in the literature [8,15,41]. This phenomenon occurs because of the previously mentioned peritectic reaction and according to the difference in Zr concentration [9,42]. According to nucleation theory, the higher the solute content, the higher the chemical driving force for nucleation, and, consequently, the larger the number density of the smaller dispersoids.

Figure 5e shows a representative volumetric diameter distribution of dispersoids from TEM, together with the fitted log-normal size distribution. For TEM data evaluation, overlapping particles were not considered. However, even when considering the overlapped ones, the trend of the size distribution for each sample was still the same, and there was no significant variation. However, the number of dispersoids that can be considered by TEM is low compared to the other techniques [42,43]. Furthermore, the dispersoids have different sizes at different locations [44,45,46]. Hence, if we consider only the middle sections of the grains, we will hit smaller dispersoids, which may lead to some deviations in the outcome from the other methods.

Figure 6 shows the comparison of the volume-normalized size distributions from all four techniques for six samples from different positions of the ingot cross-section (see Figure 1). Comparing all four methods, we find that the average diameter of the dispersoids is very similar overall for SAXS, SANS, and TEM, which is considerably smaller than for SEM. Moreover, the width of the distributions from the three former techniques are comparable, while again, SEM shows a considerably broader distribution. These differences will be discussed more in detail in Section 4.

## 4. Discussion

As shown in the literature, is well-known that the size distributions of dispersoids in 7xxx alloys are broad, which is mainly attributed to their heterogeneous distribution within the grains. Therefore, results from different techniques have to be interpreted carefully, and their quantitative comparison is only possible with limitations. This is clearly demonstrated in Figure 6, where volumetric size distributions from all four techniques employed in this work, i.e., SANS, SAXS, TEM, and SEM, are compared. The first obvious difference is that the SEM results clearly deviate from those of the three other techniques for all investigated samples. The maximum of the distribution is shifted to larger particles by almost a factor of two, and the distribution is systematically much broader. The agreement between the three other techniques (i.e., SANS, SAXS, and TEM) is reasonable, although some systematic differences also show up there. Next, we discuss each of the techniques with regard to its advantages and disadvantages, and propose possible explanations for the differences in the obtained size distributions.

Besides the limitations of the resolutions of all the techniques (discussed below), an important factor is the sampled volume. The average grain size of the samples in this work is in the order of 30–90 microns, with a tendency to be larger toward the center of the ingot. The typical areas covered by the TEM and SEM images are just a few µm^2^ and around 100 × 100 µm^2^, respectively. This means that, with TEM, only a very small fraction of a grain was covered, while an SEM image typically enclosed a few single grains at its maximum. In contrast, SAXS, in this work, typically sampled around 1000 grains, and SANS covered almost one million grains (the illuminated sample volume was given by the square of the beam size times the sample thickness). Therefore, in terms of statistical accuracy, not only regarding the number of precipitates contributing to the result, but also regarding the number of covered grains, SAXS and, particularly, SANS are preferred. In contrast, TEM is particularly sensitive to local variations/distortions of the microstructure, due to its very local character being limited to only a very small number of dispersoids. For instance, the maximum of the TEM size distribution in Figure 6a is at *D* > 30 nm, clearly deviating from the SANS/SAXS results, while for all of the other samples (Figure 6b–f), it is around 20–25 nm, in quite good agreement with SANS/SAXS. We interpret this “outlier” for the TEM size distribution in Figure 6a to be due to some local variability of the microstructure and perhaps also because the images were taken closer to a grain boundary, compared to the other samples.

The deviation of the SEM size distributions can mainly be understood in terms of resolution. With the present setup, the lower boundary for the precipitate size is around 25 nm. This value actually corresponds roughly to the maximum in the size distributions from SANS/SAXS/TEM, but the observed maximum of the SEM distribution is clearly shifted to larger sizes. The reason for not simply having a cutoff at around 25–30 nm is twofold, as described in the following:

Firstly, we speculate that the probability of detecting small particles close to the resolution limit of SEM may be strongly reduced due to contrast reasons. Secondly, in the volumetric size distribution, larger particles are weighted much stronger, thus shifting the maximum of the originally strongly asymmetric number size distribution clearly toward larger sizes. Indeed, while we observed a strongly asymmetric number distribution in Figure 4, the corresponding volumetric distribution in Figure 6d is more symmetrical and strongly shifted to larger diameters. The SEM data consistently show a considerable volume of dispersoids with diameter larger than 50 nm; this in contrast to the other techniques, which all cease at about this value. One important shortcoming of SEM in this respect could be that clusters of dispersoids (as seen, for instance, in the TEM images in Figure 5b,d) might appear as single large precipitates due to the insufficient resolution. Few large precipitates would, of course, influence the volumetric size distribution at large sizes much more strongly than the number distribution.

These shortcomings of SEM go along with the fact that the other three methods have resolution limitations regarding large particles. For TEM, this is not a fundamental restriction; rather, it is due to the fact that, in this work, we have only analyzed dispersoids that are far away from grain boundaries. These are known to be systematically smaller compared to dispersoids close to grain boundaries (see Appendix A), and, therefore, the considerably narrower size distribution with a smaller average dispersoid size for TEM is reasonable. For SANS and SAXS, however, the resolution is, by itself, strongly limited regarding large diameters. As a rule of thumb, when estimating for an upper detectable particle diameter, we can simply take the inverse of the minimum *q*-value of the experimental setup, i.e., $d_{max}=\pi /q_{min}$ [30]. This leads to an upper limit of about 50 nm for the SANS-derived data and an even smaller value (≈35 nm) for SAXS, considering the present experimental configurations. Therefore, although these limits are not sharp and depend on many details of the models used for the SANS/SAXS data analysis, diameter values larger than 50 nm for the SANS data and 35 nm for the SAXS data are considered highly unreliable. This fact may also explain the systematic difference between SANS and SAXS, the latter systematically showing a somewhat broader tail for the size distributions toward larger diameters compared to SANS. We speculate that the separation of the scattering contributions at very small *q*-values (see Appendix A) does not work as reliably for SAXS as it does for SANS, thus leading to an erroneous “broadening” of the size distributions from SAXS. However, of course, some other factors may also influence the results, e.g., scattering contributions from minor phases that may have quite a different contrast for SAXS and SANS. Finally, it also needs to be mentioned that the SANS/SAXS model applied here assumes dilute particles; this takes only the dispersoid form factor into account, but not the mutual interaction between dispersoids due to their closer packing. The clustering of dispersoids such as those seen in some of the TEM images might additionally complicate the SANS/SAXS data analysis, but this goes far beyond the scope of the present work.

## 5. Conclusions

In conclusion, we have determined the dispersoid diameter distributions in an industrial Al–Zn–Mg–Cu alloy using four different methods. The results from SANS, SAXS, and TEM agree reasonably well, but the SEM distributions deviate quite strongly, with a clear shift of the maximum to larger sizes and a considerably broader distribution. On the one hand, there is an obvious limitation of SEM, with respect to resolution, regarding small particles below about 25-30 nm, which becomes particularly pronounced due to the chosen presentation of the data as volumetric distributions. On the other hand, SANS and SAXS are resolution-limited toward large particle diameters exceeding about 50 nm for the experimental configurations employed in this work. TEM has no such principal restrictions, but suffers from the extremely local sampling, which, together with the heterogeneous distribution of the dispersoids within the grains and the large effort required for the TEM sample preparation, also causes severe limitations in this technique. Obviously, none of the techniques employed in this work is “uniquely” suited for such complex systems, where the volume fractions are low, the size distributions are broad, and the spatial distribution of the precipitates is inhomogeneous. Hence, the combination of different techniques, rather than one single technique, is necessary to cover the full complexity of such systems.

The different sensitivities of X-rays and neutrons to specific chemical elements have proven that, in the medium size regime (≈10–50 nm), dispersoids dominantly contribute to the scattering signal discussed in this work. This allowed for the derivation of size distributions from large sample volumes, which were found to be similar for SAXS and SANS, and also largely agreed with the results from TEM. After demonstrating that SAXS and SANS are suitable techniques for characterizing low-volume fractions of dispersoids in complex industrial Al alloys, this opens new possibilities for, e.g., in situ studies of dispersoid formation during thermal treatment.

## Figures and Tables

**Figure 1 materials-16-01213-f001:**
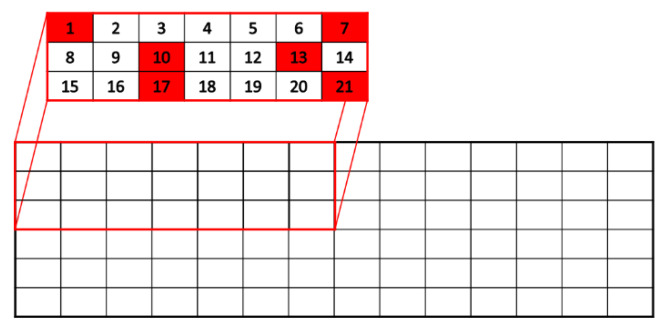
Sketch of the cross-section of the as-cast bar (width 1.5 m and height 0.45 m), indicating the location of the samples studied with SAXS and SANS within one-quarter of the cross-section (sample numbers from #1 to #21). The six samples highlighted with red color were additionally investigated with SEM and TEM.

**Figure 2 materials-16-01213-f002:**
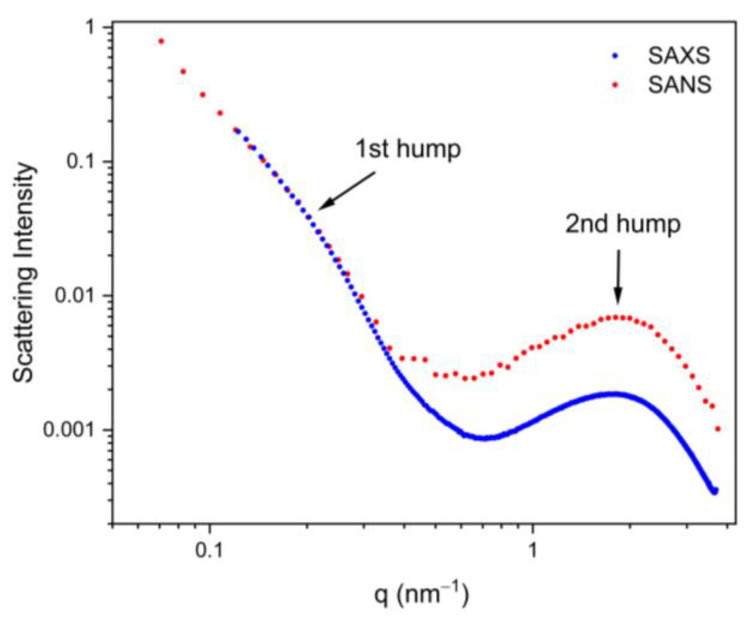
Representative SAXS (blue) and SANS (red) curves of a homogenized and water quenched sample. The SAXS curve was multiplied with a constant factor, such that the curves overlapped at low/intermediate *q*.

**Figure 3 materials-16-01213-f003:**
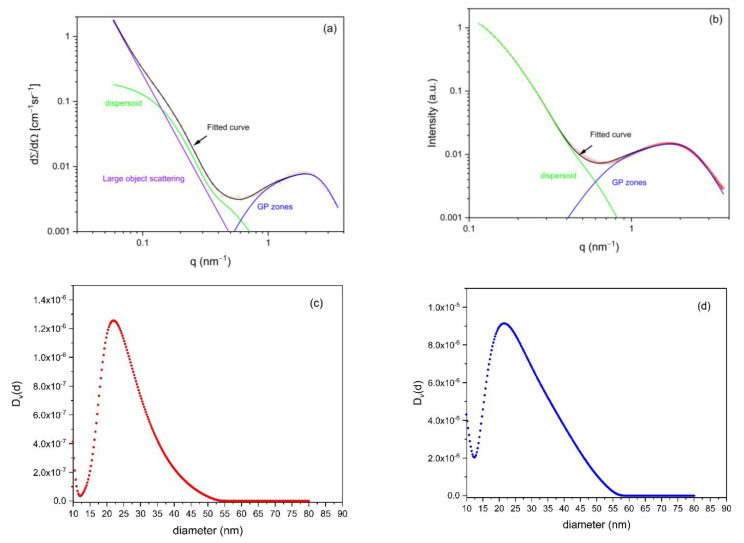
Fitted SAS curves from sample #13 with the contribution of different precipitate phases to the curve for SANS (**a**) and SAXS (**b**). Corresponding dispersoid-size distributions are shown for SANS (**c**) and SAXS (**d**), with D(d) being proportional to the volume.

**Figure 4 materials-16-01213-f004:**
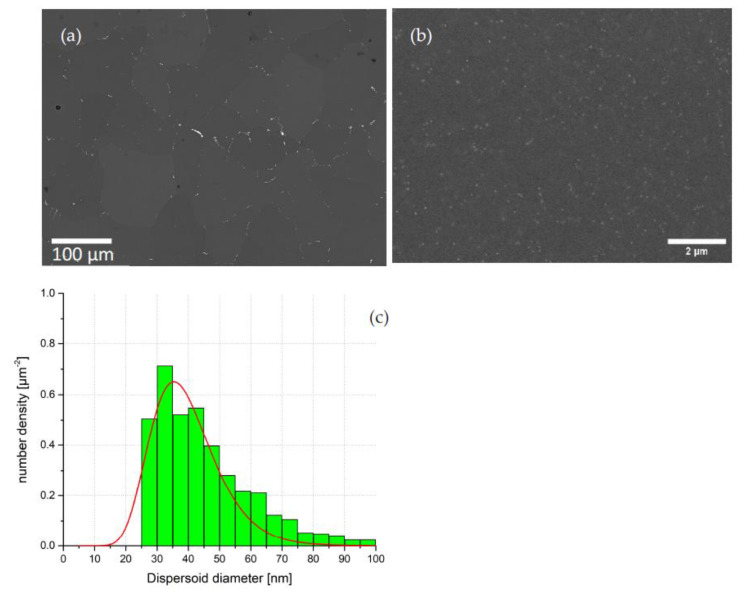
(**a**) SEM micrograph of the homogenized and water quenched sample #13. (**b**) SEM image in larger magnification showing the dispersoid distribution within a grain from (**a**). (**c**) Size distribution of dispersoids in sample #13 calculated from the SEM results.

**Figure 5 materials-16-01213-f005:**
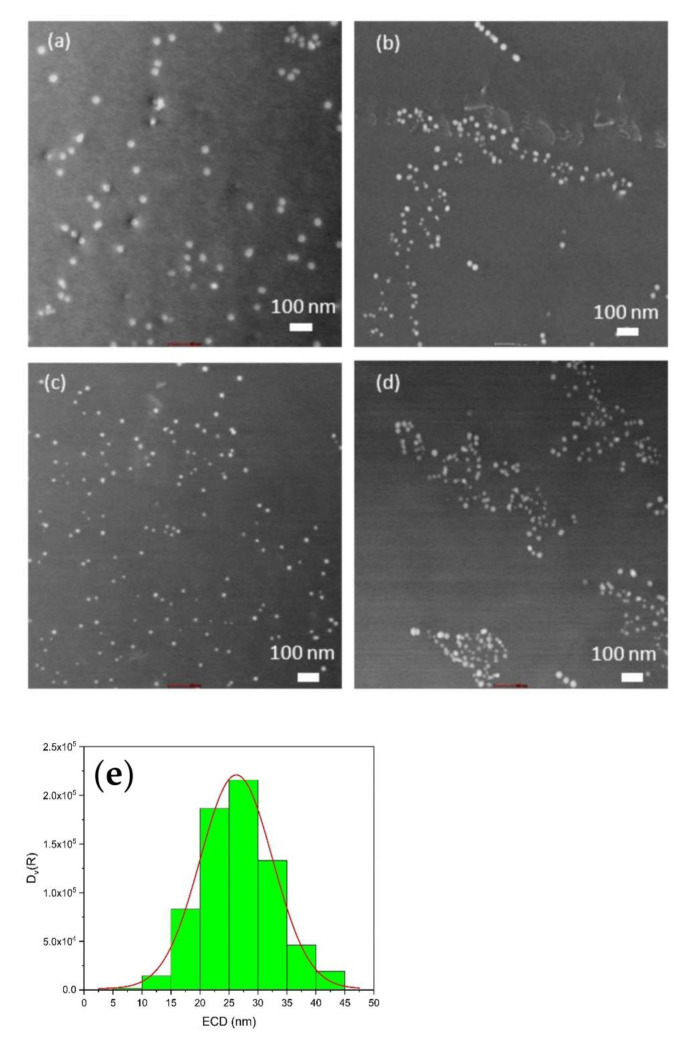
Dark field HAADF-STEM image of Zr-containing dispersoids in samples (**a**) #1, (**b**) #7, (**c**) #10, and (**d**) #17, and (**e**) size distribution of dispersoids derived from TEM.

**Figure 6 materials-16-01213-f006:**
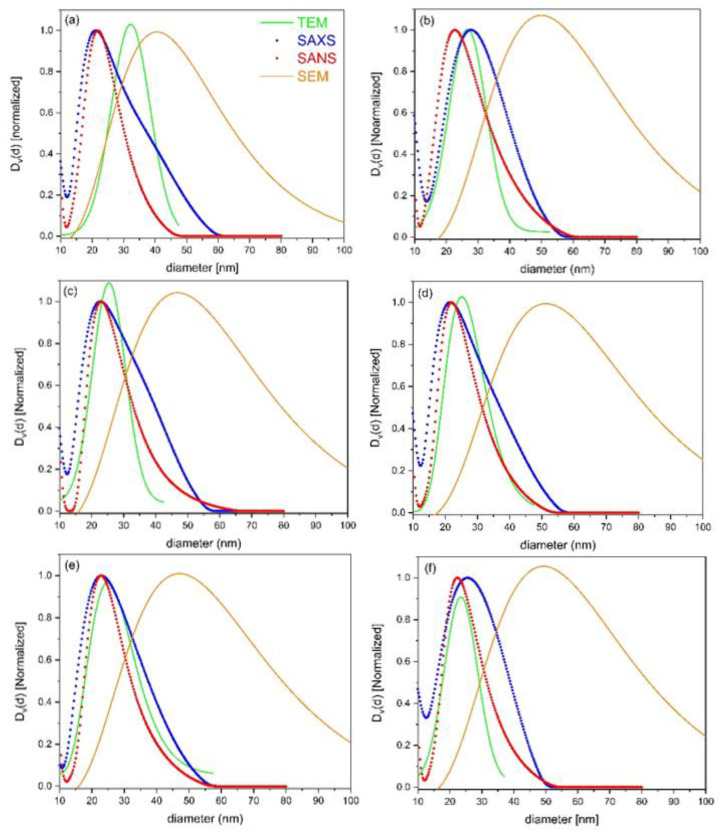
Comparison of the volume-normalized size distributions derived from SANS (red circles), SAXS (blue circles), TEM (green lines), and SEM (orange lines) for six different samples. (**a**) Sample #1, (**b**) sample #7, (**c**) sample #10, (**d**) sample #13, (**e**) sample #17, and (**f**) sample #21.

**Table 1 materials-16-01213-t001:** Chemical composition of the DC-cast Zr-containing Al-Zn-Mg-Cu alloy (wt.%).

	Al	Zn	Mg	Cu	Zr	Ti	Fe	Si
ICP OES	Balance	6.35 ± 0.25	2.07 ± 0.1	2.09 ± 0.13	0.098 ± 0.005	0.037 ± 0.006	0.029 ± 0.001	0.025 ± 0.005
XRF	Balance	6.05 ± 0.19	2.50 ± 0.07	2.25 ± 0.14	0.096 ± 0.003	0.037 ± 0.006	0.038 ± 0.008	0.032 ± 0.010

## Data Availability

The raw/processed data required to reproduce these findings cannot be shared, as the data also form part of an ongoing study.

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
