# Peer review of "Characterization of Zr-Containing Dispersoids in Al–Zn–Mg–Cu Alloys by Small-Angle Scattering"

_materials, 2023, doi:10.3390/ma16031213_

Round 1
Reviewer 1 Report
Honaramooz and co-workers reported the title of “Characterization of Zr-containing dispersoids in Al-Zn-Mg-Cu alloys by Small-Angle Scattering”, which the experimental is carefully conducted, and the results have been presented correctly, and the contents fall well into the scope of the journal. I recommend the publication of this paper after minor revision :
1. Briefly describe the advantages and disadvantages of the four characterization methods (SAXS, SANS, SEM and TEM).
2. Figure 6 in each small picture is too crowded resulting in poor clarity, please make appropriate changes.
Author Response
Reviewer comment: Honaramooz and co-workers reported the title of “Characterization of Zr-containing dispersoids in Al-Zn-Mg-Cu alloys by Small-Angle Scattering”, which the experimental is carefully conducted, and the results have been presented correctly, and the contents fall well into the scope of the journal. I recommend the publication of this paper after minor revision:
Our reply: We thank the reviewer for the constructive comments and suggestions. In the following we reply point by point to the reviewer’s recommendations. All changes in the revised manuscript are marked with red color.
Reviewer Comment 1: Briefly describe the advantages and disadvantages of the four characterization methods (SAXS, SANS, SEM and TEM).
Our reply: we have now added some more information of the advantages and disadvantages of the four used techniques in the introduction and also in the conclusion section. It is one of the most important conclusions of the present work that none of the techniques can be seen as “better” or even “unique”, hence only the combination of different techniques leads a full picture of such complex systems.
Reviewer Comment 2: Figure 6 in each small picture is too crowded resulting in poor clarity, please make appropriate changes.
Our reply: we agree with the referee that Figure 6 is quite crowded, but we would like to keep all 6 panels since it appears important to show that our results and conclusions do not only hold for one sample, but are consistent for the whole industrial ingot. Yet, we made some changes in the Figure 6 (2 columns and 3 rows instead of 3 columns and 2 rows; removal of the insets) with the hope that they are now more clear.
Reviewer 2 Report
Extensive editing of English is required
Author Response
Rewiewer comment: Extensive editing of English is required
Our reply: the manuscript has been thoroughly edited by a professional (paid) editing service provided by MDPI. We hope that the manuscript is now better readable.
Reviewer 3 Report
The authors proposed a work on different possibilities to characterize dispersoids in Al alloys. The topic is relevant for the journal and it well written. However, there is a number of issues that require clarification, and should be addressed very carefully before it can be considered for publication. Please see the attached PDF file to this review for the improvement comments.

Author Response
Reviewer #3:
Reviewer comment: the authors proposed a work on different possibilities to characterize dispersoids in Al alloys. The topic is relevant for the journal and it well written. However, there is a number of issues that require clarification, and should be addressed very carefully before it can be considered for publication. Please see the attached PDF file to this review for the improvement comments.
We thank the reviewer for the constructive comments and suggestions, and for the advice on improving the readability of the manuscript. In the following we reply point by point to the reviewer’s remarks in the PDF file. All changes in the revised manuscript are marked with red color.
Reviewer remark 1: one contextualization sentence to show the relevance of the study
Our reply: we have now added a corresponding sentence at the beginning of the abstract
Reviewer remark 2: please merge these references: [3-5]. Check throughout the paper
Our reply: all references have been checked and merged where necessary, and additionally updated according to the journal’s style.
Reviewer remark 3: the state of the art review presents good refences on the topic, but what is the novelty of the tested techniques? what did previous research allowed to conclude on the different dispersoid characterization methods? these issues should be discussed in more detail to substantiate the proposed work
Our reply: We think that one “novelty” of this paper is that SAXS and SANS are well suited complementary techniques for the quantitative characterization of precipitates not only for binary systems, but also for more complex industrial alloys. These techniques are known since many decades, but nowadays are seldom employed for metal alloys, particularly if ternary or quarternary systems are concerned. This might be due to the fact that no real space images but only intensities in reciprocal space are available, and quite some effort in terms of data modelling and evaluation is necessary to extract the information from the scattering curves. We have now added a few sentences in the introduction to stress the complementary information from small angle scattering, particularly their strength of covering macroscopic sample volumes with thousands of grains and billions of precipitates. We hope this will dispel the concerns of the reviewer to a certain extent.
Reviewer remark 4: new paragraph since these are the paper objectives
Our reply: Done
Reviewer remark 5: the name of the alloy (7050) should be also in the first line of this section
Our reply: Since the industrial name of the alloy is not relevant here, we have now replaced the term 7050 or 7xxx by Al-Zn-Mg-Cu throughout the manuscript.
Reviewer remark 6: “D33 instrument” Please clarify
Our reply: the full name of the instrument has been added to the manuscript
Reviewer remark 7: which is the justification behind the different process parameters and procedures? are these the optimal parameters? if necessary include references
Our reply: the procedures applied here are standard procedures in the (semi-automated) evaluation of TEM images. The lower limit of 15 nm2 (corresponding to a precipitate diameter of about 4 nm) is believed to be considerably below the smallest dispersoids, and was introduced in order to avoid artifacts from fluctuations. Moreover, the circularity parameter of 0.85 was applied to avoid artifacts in the size distribution from overlapping particles, and was chosen based on the fact that the dispersoids are almost perfectly spherical in shape.
Reviewer remark 8: how did the authors reach to this conclusion? “the shoulder at intermediate q (at about 0.2 nm-1) is attributed to Zr-containing dispersoids.
Our reply: this is perhaps a misunderstanding due to a linguistic inaccuracy: the detailed explanation why we come to this conclusion was explained in detail in the following paragraph. To make this clearer, we have added the half sentence "....as explained and justified below."
Reviewer remark 9: a table can be introduced to quantify the differences between evaluation methods for the different samples, to facilitate comparison
Our reply: this is exactly what Figure 6, in which the volume-normalized size distributions of SAXS, SANS, TEM and SEM were compared, is supposed to do.
Reviewer remark 10: please clarify this sentence. “The reason for not having simply a cutoff at around 25-30 nm, with the highest volume of precipitates at these values may be twofold”.
Our reply: Similar to the point 8. above, this may be a misunderstanding due to a linguistic inaccuracy: the explanation is added immediately in the following paragraph. Again, we have added a half sentence "....as described in the following."
Round 2
Reviewer 2 Report
Accept it in present form
Reviewer 3 Report
Thank you for the improvements